# Log-Lattices for Atmospheric Flows

**Quentin Pikeroen , Amaury Barral** , **Guillaume Costa** and **Bérengère Dubrulle** *

University Paris-Saclay, CEA, CNRS, SPEC, 91191 Gif-sur-Yvette, France; quentin.pikeroen@lsce.ipsl.fr (Q.P.); guillaume.costa@cea.fr (G.C.)

*   Correspondence: berengere.dubrulle@cea.fr

**Abstract:** We discuss how the projection of geophysical equations of motion onto an exponential grid allows the determination of realistic values of parameters at a moderate cost. This allows us to perform many simulations over a wide range of parameters, thereby leading to general scaling laws of transport efficiency that can then be used to parametrize the turbulent transport in general climate models for Earth or other planets. We illustrate this process using the equation describing heat transport in a dry atmosphere to obtain the scaling laws for the onset of convection as a function of rotation. We confirm the theoretical scaling of the critical Rayleigh number, $\mathrm{Ra}_c \sim E^{-4/3}$, over a wide range of parameters. We have also demonstrated the existence of two regimes of convection: one laminar regime extending near the convection onset, and one turbulent regime occurring as soon as the vertical Reynolds number reaches a value of $10^4$. We derive general scaling laws for these two regimes, both for the transport of heat and the dissipation of kinetic energy, and values of anisotropy and temperature fluctuations.

**Keywords:** convection; rotation; turbulence




## 1. Foreword by B. Dubrulle

I met Jack in 1999, when I came to the MMM division of NCAR for a one-year sabbatical. I had been attracted there by Jack's reputation from conversations with Annick Pouquet, Uriel Frisch, and Maurice Meneguzzi. Being a theoretician of turbulence interested in geophysical applications, I then knew that I would find in Jack a very good interlocutor and benefit greatly from his physical insights, his broad knowledge about turbulence and geophysical flows, and his open mind. I met Jack regularly during my stay, and we spoke about all sorts of topics. It was after a discussion with them that I started to investigate Rayleigh–Bénard flows and heat transfer properties—I published two papers on this topic that year. About Jack, I keep the memory of a true "gentleman of science", very kind to junior scientists (as I still was at that time), with a great sense of humor and an immense knowledge that he was keen to share. It is certainly thanks to Jack that I jumped into the modeling of geophysical flows, and I will always be grateful to him for this.

## 2. Introduction

Ultra-high Reynolds number flows are ubiquitous in geosciences due to small viscosities, large dimensions or velocity. They are described by the Navier–Stokes equations (NSEs). A natural control parameter of the NSEs is the Reynolds number $Re = LU/\nu$, built using the viscosity $\nu$ and characteristic length $L$ and velocity $U$. Classical turbulent flows are thought to be described by NSEs, with $Re \gg 1$. In 1941, Kolmogorov [K41] used such equations to predict the shape of the energy spectrum $E(k)$ derived from the Fourier transform of the velocity correlation function for isotropic and homogeneous turbulence stirred at a constant rate $\epsilon$. He found that it should scale like $E(k) \sim \epsilon^{2/3}k^{-5/3}$ in the range $1/L \ll k \ll \nu^{-3/4}\epsilon^{1/4}$, where $k$ is the wavenumber. This prediction was verified in 1962 on data from a turbulent flow in Seymour Narrows [1] and appears to be one of the most robust laws of turbulence [2,3], being independent of the boundaries or the stirring process.

At large wavenumbers, viscous processes take over, and the spectrum decays very fast, so that the energy contained in wavenumbers greater than $k_d = \nu^{-3/4}\epsilon^{1/4}$ is negligible. The overall behavior of $E(k)$ can be used to infer the typical number of dynamically active modes as $N = (k_d L)^3$. In geophysical flows, such a number can be quite large; for example, for the atmosphere, $L = 10^3$ km (the size of the large typhoons or cyclones), while $k_d^{-1}$ is smaller than 10 mm, resulting in $k_d L > 10^8$. Direct numerical simulations of the Navier–Stokes equations for such flows are thus impossible, as the total computational cost of reading-in/writing-out and coupling all these modes exceeds the capability of current most powerful computers by many orders of magnitude.

If one wishes to simulate ultra-large Reynolds numbers like the atmosphere, one has no other way out than to empirically decimate modes via a clever selection of grid points or modes. Simulating viscous flows with just as many scales as needed to "get the physics right" has been and still is the holy grail of all researchers in the computational fluid community. If a well-established theory of turbulence was available, including a deep understanding of all interactions between scales, the quest would probably be over by now. In the absence of such a complete theory, we need empirical yet clever strategies for mode number reduction. Jack Herring worked many years on such an issue, using, e.g., two-point closure (see, e.g., ref. [4]).

Nowadays, the most popular approach is to use a large Eddy simulation strategy, in which only large scales are simulated. Present climate models have a grid size of 10 km, allowing the handling of data volumes in 2 CPU seconds per time step. However, there is no free lunch; the price to pay for such a mode reduction is the addition of a (sometime very large) damping to avoid accumulation of energy at the smallest simulated scale. In LES-type climate simulations, the damping is the same as if the atmosphere was made of peanut butter and the ocean of honey, so that no fluctuations can develop. This is problematic to capture possible bifurcations, such as those observed in von Karman flows [5].

In this paper, we consider another class of model in which mode reduction is achieved by keeping modes following a geometric progression. Such an approximation leaves out a lot of possible interactions, as we shall see. However, it allows for reaching very small scales with a very small number of modes. In the atmosphere, for example, only 27 modes are necessary to go from $k_0 = 1/L$ to $10^8/L$ by a geometrical progression of step 2 (38 with a step being the golden number). This means that we can perform a "resolved" 3D simulation of an atmospheric flow with less than $2 \times 10^4$ modes (respectively, less than $6 \times 10^4$), corresponding to the number of modes involved in an LES simulation at a resolution of 37 km. The corresponding models correspond to fluid dynamics on log-lattices, the properties of which were detailed in [6]. For example, they respect the classical and basic properties of the Navier–Stokes equation, such as the constancy of the energy flux in the inertial range. Given the potential of such models to describe ultra-high Reynolds number flow at a low price, we investigate here further properties of such models with respect to one important open problem of atmospheric flows, which was dear to Jack, namely the influence of rotation on convection.

### 3. Log-Lattice Framework

Consider a velocity field that obeys the Navier–Stokes equation (NSE), with viscosity $\nu$ and forcing. Its Fourier transform, denoted $u(t, k)$, is a complex field that obeys the equation:

$$\mathbf{i}k_j u_j = 0,$$
$$\partial_t u_i + \mathbf{i}k_j u_i * u_j = -\mathbf{i}k_i p - \nu k^2 u_i + f_i, \tag{1}$$
$$u(k, t) = u^*(-k, t).$$

where $*$ denotes the traditional convolution product, that involves coupling of modes $u(t, p)$ and $u(t, q)$ for each $k$, such that $k_i = p_i + q_i$ for any $i$th component. The (constant) density has been set to 1 for convenience, and thus disappears in front of the pressure term

in Equation (1). The convolution sum is computed directly, but is less costly than a fast Fourier transform because only a limited amount of local interactions are kept [6].

In traditional spectral simulations of the NSE, the wavenumbers are discretized on a linear grid, and can be written as $k = (m_1, \ldots, m_3)$ with $m_i \in \mathbb{Z}^3$. The convolution is ensured by the condition $m = n + q$ for any $(m, n, q) \in \mathbb{Z}^3$, this allows for a lot of interactions between Fourier modes.

In the log-lattice framework, we now impose that the wavenumbers follow a geometric progression:

$$k = (\lambda^{m_1}, \ldots, \lambda^{m_3}), \tag{2}$$

with $m_i \in \mathbb{Z}^3$, where $\lambda$ is a parameter yet to be defined. It is imposed by the condition that the convolution product appearing in Equation (1) has some non-zero solution, which is only possible if the equation:

$$\lambda^m = \lambda^p + \lambda^q, \tag{3}$$

has some solutions for any $m, p, q \in \mathbb{Z}^3$. As discussed in [6], Equation (3) can only be achieved provided $\lambda$ takes some specific values, among which $\lambda = 2$, $\lambda \equiv \phi = (1 + \sqrt{5})/2 \approx 1.618$, the golden mean and $\lambda \equiv \sigma = \approx 1.325$ (the plastic number ($\sigma$ is defined as the common real root to $\sigma^3 - \sigma - 1 = 0$ and $\sigma^5 - \sigma^4 - 1 = 0$)). Different values of $\lambda$ correspond to different numbers of coupling between modes; they become more and more numerous and less and less local as $\lambda \to 1$ [6]. In this sense, log-lattices can be seen as a special case of the sparse Fourier model [7], the REWA model [8], or fractal decimated models [9,10], in which the non-linear interactions of the NSE are projectively decreased randomly or in a scale-invariant manner.

The choice of $\lambda$ is fixed by several considerations: larger values of $\lambda$ are cheaper, both in computation time and in memory. However, lower values of $\lambda$ give rise to more interactions, and thus more fluctuations, which is more realistic for simulating a turbulent flow. As discussed in [11], the case $\lambda = 2$ should be avoided when simulating incompressible dynamics because it lacks backscatter. On the other hand, ref. [12] checked whether the scaling law properties, such as the spectral slope or blow-up exponent, do not depend on the value of $\lambda$. Therefore, we work below with $\lambda = \phi$, which is the best compromise between all constraints.

### 3.1. Energy Spectra

It is possible to write the mean energy that measures the scaling properties of the mode at $k_n = \lambda^n$ as:

$$E_m(n) = <||u(k)||^2 >_{S_n}, \tag{4}$$

where the average is taken over wavenumbers in the shell delimited by spheres with radii $\lambda^{n-1}$ and $\lambda^n$. Specifically:

$$<||u(k)||^2 >_{S_n} = \frac{1}{N_k} \sum_{\lambda^{n-1} \leq ||k|| < \lambda^n} \sum_{\alpha=1}^{d} |u_\alpha(k)|^2, \tag{5}$$

where $d$ is the space dimension and $N_k \sim (\log(k))^{d-1}$ is the number of wavenumbers in the shell $S_n$.

From this quantity, one can define a pseudo-energy spectrum as:

$$E(k) = \frac{\lambda^{-n}}{\lambda - 1} <||u(k)||^2 >_{S_n}, \tag{6}$$

Examples of such a spectrum in $d = 3$ are shown in Figure 1 for $\lambda = 2$. One sees that it is self-similar and displays a very clear $k^{-5/3}$ law. This spectrum has been obtained by simulating Equation (1) with $20^3$ modes, which can easily be performed on a PC. Due to the exponential spacing, it enables reaching resolutions and inertial ranges even larger than

those achieved by oceanic measurements, a performance which is out-of-reach of DNS simulations of the NSE. This shows the importance of FDLL in geophysics.

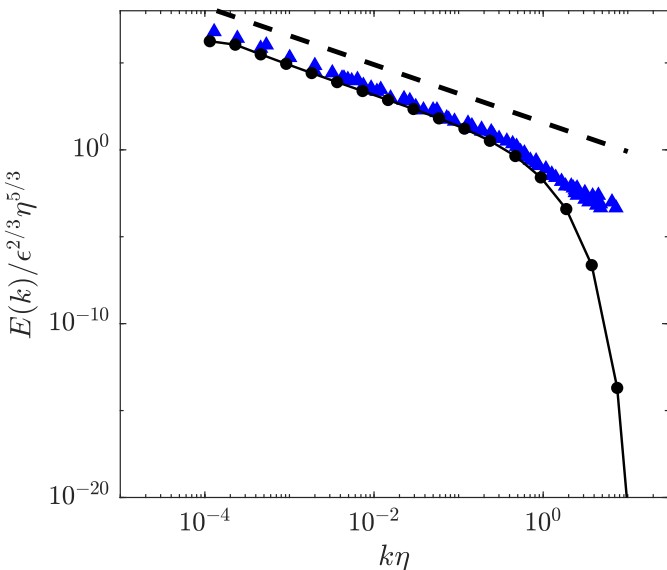

**Figure 1.** Renormalized energy spectrum $E(k)/\epsilon^{2/3}\eta^{5/3}$ as a function of the renormalized wavenumber $k\eta$ for simulation of the Navier–Stokes equation on a $20^3$ log-lattice (black dotted line) and for a turbulent flow in Seymour Narrows (Ocean) (blue squares). The black dashed line corresponds to $E(k) \sim k^{-5/3}$. This figure was drawn by the authors using data extracted from the published graph shown in [1].

### 3.2. Generalizations

Equations (1)–(3) define the Navier–Stokes equation on a log-lattice. By extension, any equation of fluid dynamics on s log-lattice can be defined by performing the following two steps [6]: (i) Write the equation in the Fourier-space. (ii) Replace any convolution product by the convolution on a log-lattice. This construction guarantees that the resulting equation obeys all the conservation laws and symmetries of the original equations [6].

### 3.3. Limitations of Log-Lattices

Computations on log-lattices involve a number of limitations. First, due to the local nature of the convolution in Fourier space, log-lattices are unable to describe the so-called "non-local" interactions that are involved, for example, in the shearing of small eddies by large eddies [13]. This may explain why log-lattice simulations do not display intermittency for the structure functions [14]. Due to their sparsity, log-lattices cannot describe dispersive wave resonances, like inertial or gravity waves. This means that they cannot capture dissipative phenomena induced by these waves, and that they can only capture the dissipation due to small-scale turbulent eddies. Finally, due to the spectral nature of the construction, it may appear that the log-lattice framework is only appropriate for homogeneous flows, i.e., far from boundaries. In a recent work, ref. [15] have, however, shown that the extension to flow with boundaries is possible via lattice symmetrization around the boundary and careful treatment of the resulting discontinuity. Despite the high relevance of this situation for geophysical flows, we here concentrate on the simpler case of homogenous flows, and show how log-lattice simulations enable the recovery of some well-known features of homogeneous rotating convections.

## 4. Homogeneous Rotating Convections on Log-Lattices

### 4.1. Definitions

We now consider a rotating homogeneous fluid, with coefficient of thermal dilation $\alpha$, viscosity $\nu$ and diffusivity $\kappa$, subject to a temperature gradient $\Delta T$ over a length $H$ and vertical gravity $g$. Its dynamics are given by the HRB set of wequations [11,16–18],

$$\partial_t u + u \cdot \nabla u + \frac{1}{\rho_0}\boldsymbol{\nabla} p + 2\boldsymbol{\Omega} \times u = \nu \nabla^2 u + \alpha g \theta \vec{z} - f u, \tag{7}$$

$$\partial_t \theta + u \cdot \boldsymbol{\nabla}\theta = \kappa \nabla^2 \theta + u_z \frac{\Delta T}{H} - f\theta, \tag{8}$$

$$\nabla \cdot u = 0, \tag{9}$$

where $u$ is the velocity, $\theta$ is the temperature deviation from the equilibrium profile (where $\theta = 0$), which means by definition that $T = -\Delta T z/H + \theta$, $\boldsymbol{\Omega}$ is the rotation vector, $\rho_0$ is the (constant) reference density, $p$ is the pressure and $f$ is a (Rayleigh) friction term, accounting for the friction at the boundary layer that cannot be resolved by the present framework. We will assume below that this friction is concentrated only on a large scale. As proven in [11], this friction is mandatory to allow the system to reach well-defined stationary states. Indeed, for periodic boundary conditions, exponential instabilities can grow because there is no wall to stop them. They are characterized by the accumulation of energy on a large scale. Introducing a large scale friction allows damping of the inverse cascade and avoids the concentration of energy on the large scale. This set of equations has to be completed with boundary conditions. In this paper, we consider periodic boundary conditions and focus on the case when the rotation is aligned with the $z$-axis, $\boldsymbol{\Omega} = \Omega e_z$.

### 4.2. Non-Dimensional Numbers

We can use five independent non-dimensional numbers to characterize the system:

- The Rayleigh number $\mathrm{Ra} = \alpha g H^3 \Delta T/(\nu \kappa)$, which characterizes the forcing by the temperature gradient.
- The Prandtl number $\mathrm{Pr} = \nu/\kappa$, which is the ratio of the fluid viscosity to its thermal diffusivity.
- The Nusselt number $\mathrm{Nu} = JH/\kappa\Delta T$ that characterizes the mean total heat flux is the $z$ direction is $J = \partial_z < u_z\theta > -\kappa\Delta T$.
- The Ekman number $\mathrm{E} = \nu/(2\Omega H^2)$, measuring the importance of the rotation with respect to the diffusive process.
- The Rossby number $\mathrm{Ro} = \sqrt{\alpha g \Delta T}/(2\Omega\sqrt{H})$, measuring the importance of the rotation with respect to buoyancy. In terms of other variables, we have $\mathrm{Ro} = \mathrm{E}\sqrt{\mathrm{Ra}}/(\sqrt{Pr})$.
- The friction coefficient $F = f\sqrt{H}/\sqrt{\alpha g \Delta T}$, which provides the intensity of the Rayleigh damping.

### 4.3. Equations on Log-Lattice

The regimes we want to explore are very turbulent regimes where the viscosity and diffusivity do not play any role anymore. Therefore, it is natural to adimensionalize the equation in terms of "inertial quantities", i.e., using the vertical width $H$ as a unit of length, the free fall velocity $U = \sqrt{\alpha g \Delta T H}$ as a unit of velocity, and $\Delta T$ as a unit of temperature.

Then, to define the Rotating Homogeneous Rayleigh–Benard (RHRB) equations on log-lattices, we take the Fourier transform of Equation (7) that can be written in non-dimensional form as (with the Einstein convention on summed repeated indices):

$$\mathbf{i}k_j u_j = 0, \tag{10}$$

$$\partial_t u_i + \mathbf{i}k_j u_i * u_j = -\mathbf{i}k_i p + \theta\delta_{i3} - Fu_i\delta_{k\approx k_{\min}}$$

$$-\sqrt{\frac{\mathrm{Pr}}{\mathrm{Ra}}}k^2 u_i - \frac{1}{\mathrm{Ro}}\epsilon_{i3k}u_k \tag{11}$$

$$\partial_t\theta + \mathbf{i}k_j\theta * u_j = u_z - \frac{k^2\theta}{\sqrt{\mathrm{Ra}\,\mathrm{Pr}}} - F\theta\delta_{k\approx k_{\min}}, \tag{12}$$

$$u(k,t) = u^*(-k,t), \tag{13}$$

$$\theta(k,t) = \theta^*(-k,t). \tag{14}$$

where the Dirac $\delta_{k\approx k_{\min}}$ filters out the small scales.

In these equations, the convolution product is taken over the log-lattice, see Equation (3).

*4.4. Convection Onset*

Convection is an instability, so it sets-up at a certain critical value of the parameter $\mathrm{Ra}_c$. Detailed computation of this parameter is performed in [19] for the case $F = 0$. In a nutshell, we assume that we are very near the threshold, so that deviations from the "equilibrium state" $u = \theta = 0$ are small. This will allow us to neglect all non-linear terms in Equation (14). Then, we look for solutions behaving as:

$$u(k,t) = (u(k), v(k), w(k))e^{\sigma t},$$
$$p(k,t) = p(k)e^{\sigma t}, \tag{15}$$
$$\theta(k,t) = \theta(k)e^{\sigma t},$$

where $\sigma$ is the growth rate of the instability; if $\sigma$ has a negative real part, then all perturbation decay, while instability develops when the real part of $\sigma > 0$. Plugging this decomposition into Equation (14) and neglecting non-linear terms, we get:

$$ik_x u + ik_y v + ik_z w = 0,$$

$$\sigma u - \frac{v}{\mathrm{Ro}} = -ik_x p - \sqrt{\frac{\mathrm{Pr}}{\mathrm{Ra}}}k^2 u,$$

$$\sigma v + \frac{u}{\mathrm{Ro}} = -ik_y p - \sqrt{\frac{\mathrm{Pr}}{\mathrm{Ra}}}k^2 v, \tag{16}$$

$$\sigma w = -ik_z p - \sqrt{\frac{\mathrm{Pr}}{\mathrm{Ra}}}k^2 w + \theta,$$

$$\sigma\theta = w - \frac{1}{\sqrt{\mathrm{Ra}\,\mathrm{Pr}}}k^2\theta.$$

This represents a linear, homogeneous system of equations in the variable $(u, v, w, p, \theta)$. If we want this system to have other solutions than $(0, 0, 0, 0, 0)$, we must impose the determinant of the system to be zero, which provides us with an expression linking $\sigma$, $k$ and the parameters of the system. Taking $Pr = 1$ for simplicity (the general case leads to the same conclusions with different prefactors), we get:

$$\sigma^2 + 2k^2\,\mathrm{Ra}^{-1/2}\,\sigma + \frac{k^4}{\mathrm{Ra}} + (1+\mu^2)^{-1}(\frac{\mu^2}{\mathrm{Ro}^2} - 1) = 0, \tag{17}$$

where $\mu^2 = k_z^2/(k_x^2 + k_y^2)$. This is a second order equation for $\sigma$, there are therefore two solutions. Since the prefactor of $\sigma$ is positive, it means that the sum of the two solutions is negative. To ensure that there exists at least one solution with a positive real part, we must then ensure that the product of the solution, given by a term independent of $\sigma$, is negative. To study the consequences of this condition, we consider two limiting cases.

Onset at Zero Rotation

We first consider the case with no rotation, $Ro \to \infty$. Then, the condition for instability is $\frac{(1+\mu^2)k^4}{Ra} - 1 < 0$. Since $(1+\mu^2)k^4$ is at a minimum when $k_{min} = (2\pi, 2\pi, 2\pi)$, where $k_{min}$ is the minimal wavenumber, this is achieved whenever $Ra > \frac{3}{2}k_{min}^4$. In our simulations, $k_{min} \sim 2\pi\sqrt{3}$; this gives $Ra_c \sim 2.1 \times 10^4$.

*4.5. Onset at Large Rotation*

In the limit of large rotation, $Ro \to 0$ and the condition for instability now reads $\frac{k^4}{Ra} + (1+\mu^2)^{-1}(\frac{\mu^2}{Ro^2} - 1) < 0$. Introducing $\alpha = k^4/Ra$, we may further simplify the condition by noting that at the instability threshold, $\mu$ is close to 1, and $k_z^2 \ll k_x^2 + k_y^2 \sim Ra$. The condition for instability then becomes $Ra^{3/2} > 4\pi^2 E^{-2} \frac{1}{(1-\alpha)\sqrt{\alpha}}$. The instability criterion is given by the condition that there exists at least $\alpha$ such that the equality is satisfied, i.e., when the minimum of the right-hand side of the condition of instability is achieved. This is true for $\alpha = 1/3$, resulting in $Ra^{3/2} > 4\pi^2 E^{-2} \frac{3\sqrt{3}}{2}$, giving a critical Rayleigh number

$$Ra_c = (6\pi^2\sqrt{3})^{2/3} E^{-4/3} \approx 22 E^{-4/3}. \tag{18}$$

We note that this equation is similar to Equation (184) on p. 106 of [19]. This means that the larger the rotation (the smaller E), the more difficult it is to achieve convection; rotation stabilizes the flow.

*4.6. Phenomenology When F = 0*

In the case when $F = 0$, there are a number of simple physical arguments that provide the scaling laws relation between non-dimensional numbers.

4.6.1. Non-Rotating Case

We first consider the non-rotating case, $E = \infty$. In this case, the phenomenology of homogeneous convection distinguishes three regimes for the behavior of the heat flux as a function of the forcing [11,16]:

(I): When $Ra \leq Ra_c$, we are in the laminar case. The fluid is at rest, $< u_z\theta >= 0$, and the heat flux is only piloted by the Fourier law, so that $J = \kappa\Delta T/H$ and $Nu = 1$.

(II): Above the critical threshold for instability, when $Ra >\sim Ra_c$, convection sets in, $< u_z\theta >$ starts becoming positive, and we have $Nu \sim (Ra - Ra_c)^\chi$, where $\chi$ is an exponent characterizing the (super)-critical transition to convection.

(III): When $Ra \gg Ra_c$, the turbulence becomes fully developed, and we are entering an "ultimate" regime (also called the Spiegel regime), in which the heat flux does not depend on the viscosity or the diffusivity any more. In that case, we have $Nu \sim (Ra\,Pr)^{1/2}$ [20–22] and $Re \sim (Ra\,/\,Pr)^{1/2}$ [23].

This regime is very difficult to observe in DNSs because of the need for high resolution to be able to cope with $Ra \gg Ra_c$. In the atmosphere, Ra is typically of the order of $Ra \sim 10^{18-22}$, so we expect the atmosphere to be in this ultimate regime.

Note that in the case where boundaries are present, there is the possibility of an intermediate regime (the Malkus regime), where the heat flux is piloted by the boundary layers and $Nu \sim Ra^\gamma$, $\gamma \sim 1/3$. This regime is frequently observed in laboratory experiments, but it does not apply to homogeneous turbulence because of the absence of boundaries.

4.6.2. Rotating Case

Let us consider the case with rotation. As shown by many studies, the rotation has a stabilizing influence on the flow, so that the threshold for instability now increases with increasing rotation. Detailed studies show that $Ra_c \sim E^{-4/3}$ [24]. In addition, the rotation modifies the structure of the convective cells, which become aligned with the vertical axis in the case of strong rotation. This profoundly changes the heat transfer. To account for this

effect, ref. [25] suggests performing the same phenomenology as in the non-rotating case, using $\mathrm{Ra}\,E^{4/3}$ instead of Ra. The two turbulent regimes in the presence of rotation are now:

(R1) A rotating Malkus regime in the presence of boundaries, where the heat flux is independent of the height of the cell and in which $\mathrm{Nu} \sim \mathrm{Ra}^3\,E^4$. This regime cannot be present in HRB equations.

(R2) A rotating ultimate regime, also called a geostrophic turbulent (GT) regime, at Reynolds numbers larger than the threshold for onset of turbulence in the boundary layers, or when boundaries do not limit the heat flux, like in HRB equations. This regime is then found by stating that the relation between Nu and $\mathrm{Ra}\,E^{4/3}$ and Pr should be such that the energy flux $J$ is independent of $\kappa$ and $\nu$, resulting in [25].

$$\mathrm{Nu} \sim \mathrm{Ra}^{3/2}\,E^2\,\mathrm{Pr}^{-1/2}\,. \tag{19}$$

An interesting property of the geostrophic turbulent regime is that it can be expressed as a universal law, independent of the rotation and the Prandtl number, using the "turbulent" coordinates [25]:

$$\mathrm{Nu}_* = \frac{\mathrm{Nu}\,E}{\mathrm{Pr}},$$
$$\mathrm{Ra}_* = \frac{\mathrm{Ra}\,E^2}{\mathrm{Pr}}. \tag{20}$$

In this case, the relation (19) becomes:

$$\mathrm{Nu}_* \sim \mathrm{Ra}_*^{3/2}\,. \tag{21}$$

In this regime, we further have $\mathrm{Re} \sim \mathrm{Ra}\,E^2\,/\,\mathrm{Pr}$ [23].

### 4.6.3. Log-Lattice Simulation Details

To simulate these equations, we can perform log-lattice simulations. The minimum wave vector of the grid is set to $k_{\min}^{1D} = 2\pi$ to match a simulation on a box of size $\tilde{L} = 1$. Note that the absence of wavenumbers ($k = 0$) in our simulation means that we will not be able to capture large-scale vertical structures that are often observed in rotating turbulence. However, as shown below, this does not seem to affect the scaling laws of the transport of heat and momentum that we observe. The grid size $N$ (such that the maximum wavenumber in the x, y or z direction is proportional to $\lambda^N$) is then set to reach the dissipative scale for both velocity and temperature. We have verified that the size of the grid for 3D simulations ($N \geq 13$) does not affect the mean value of the observables $\mathrm{Nu}, \mathrm{Re}, \ldots$, which are already converged for grids of size $N \geq 6$. However, the tail of the pdfs does depend on $N$. Another 3D simulation set at $N = 20$ (not shown here, both vs. Ra and Pr) displays the same scaling laws as the $N = 13$ case, confirming this analysis.

## 5. Results

### 5.1. Non-Rotating Case

As reference, we have performed simulations without rotation for various Ra values up to $10^{25}$ at $\mathrm{Pr} = 1$ and $\mathrm{Pr} = 0.7$ and for $F = 1$. Note that present direct numerical simulations for these Prandtl numbers are limited to $\mathrm{Ra} \approx 10^{15}$ [26]. The resulting heat transfer, Nu, as a function of the Rayleigh number Ra is shown in Figure 2. In that case, we observe that the convection starts as $\mathrm{Ra}_c = 6 \times 10^5$, which is larger than the value predicted by the linear theory at $F = 0$, and those observed in previous log-lattice simulations [11]. This means that friction stabilizes the convection in the non-rotating regime. In addition, we observe a near onset regime $\mathrm{Nu} \sim (\mathrm{Ra} - \mathrm{Ra}_c)^{3/2}$, which then turns into an asymptotic regime, where $\mathrm{Nu} \sim \sqrt{\mathrm{Ra}}$. This regime itself splits in two distinct regimes, characterized by the same scaling law $\mathrm{Nu} \sim \mathrm{Ra}^{1/2}$, but with a very different amplitude: (i) a *laminar*

*regime*, characterized by low fluctuations of the heat transfer. This laminar regime is well fitted by an empirical law

$$\text{Nu} = A(\text{Ra} - \text{Ra}_c)^{3/2}/(\text{Ra} / \text{Ra}_t + 1), \tag{22}$$

where $A = 7$, $\text{Ra}_c = 5 \times 10^5$ is the critical number for convection onset, and $\text{Ra}_t = 5 \times 10^6$ is the transition number from near-onset to the asymptotic regime. This laminar regime is followed by (ii) a *turbulent regime*, with much larger fluctuations. The typical time behavior of the heat transfer in the two regimes is displayed in Figure 3, which showcases both regimes. The laminar regime is stable up to $\text{Ra} \approx 10^{12}$, and unstable above; if we wait long enough, the solution jumps from the laminar regime to the turbulent regime, as illustrated in Figure 3. Looking at Nu as a function of time in this regime in Figure 3, we observe that the simulation first follows a rather long evolution along the laminar regime, before suddenly transitioning towards the turbulent regime as instability develops, increasing its energy by several orders of magnitude in the process.

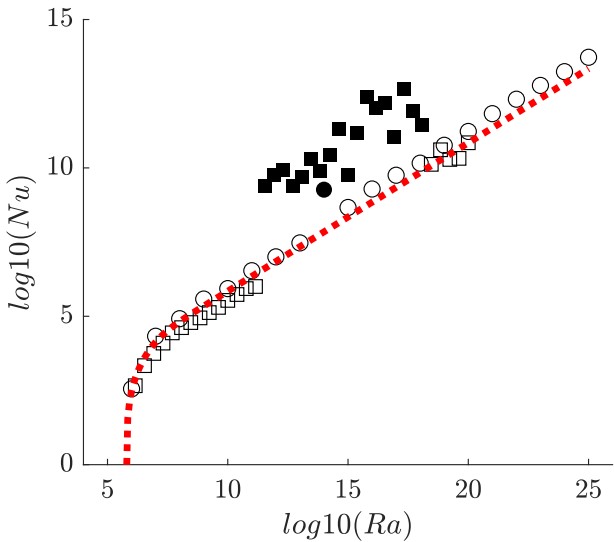

**Figure 2.** Non-dimensional heat transfer Nu vs. Rayleigh number Ra in 3D for Pr = 0.7 (circle) and Pr = 1 (squares). The dotted line corresponds to the empirical law: $\text{Nu} = 7(\text{Ra} - \text{Ra}_c)^{3/2}/(\text{Ra} / \text{Ra}_t + 1)$, with $\text{Ra}_c = 6 \times 10^5$ and $\text{Ra}_t = 5 \times 10^6$ that connects the near-convection onset regime to the asymptotic law $\text{Nu} \sim 20\sqrt{\text{Ra}}$ for large Ra, corresponding to asymptotic non-rotating ultimate regime scaling. This regime is itself split into a laminar regime (open symbols) and a turbulent regime (filled symbols).

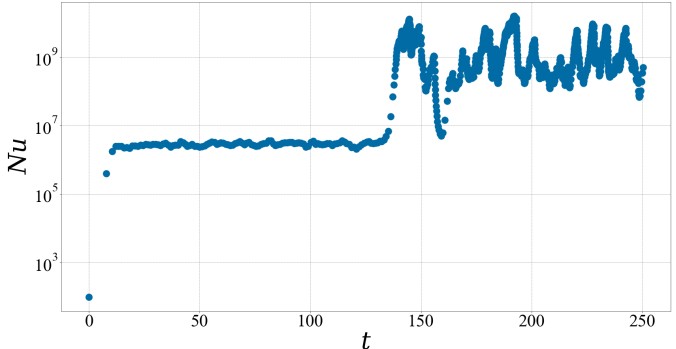

**Figure 3.** Non-dimensional heat transfer Nu versus time $t$ for $\text{Ra} = 3.46 \times 10^{11}$. The flow starts in a laminar regime, then abruptly transitions to a turbulent regime.

### 5.2. Rotating Case

Parameter Space and Critical Rayleigh Number

We turn now to the case with rotation. Figure 4 shows the parameter space we have explored, with E ranging from $10^{-9}$ to $10^{-1}$ and Ra up to $10^{14}$. In this range of parameters, we observe typically three types of behaviors: (i) Conductive behaviors, where both the velocity and temperature fluctuations are zero and Nu = 1. These cases are shown as white symbols in Figure 4. (ii) Transitional regimes, where the velocity and temperature fluctuations are decaying very slowly but steadily along with Nu over the time of the simulation, so that asymptotically, they are likely to converge to the conductive limit. The typical timescale to reach this limit increases as E decays and Ra increases. To save computational time, we stopped the simulation before reaching the limit for $E \geq 10^{-6}$, but have reported these points as yellow points in Figure 4. (iii) Convective regimes, where both the velocity and RMS velocities reach a stationary state; these points are reported as black points in Figure 4. This representation allows for a clear view of the stabilizing influence of rotation on the convection threshold. In the asymptotic regime of large rotation, $E \ll 1$, such an influence is very well described by the prediction of the linear theory at $F = 0$, Equation (18), which is reported as a red line in the diagram. This means that, as rotation increases, the friction becomes less important in the dynamics.

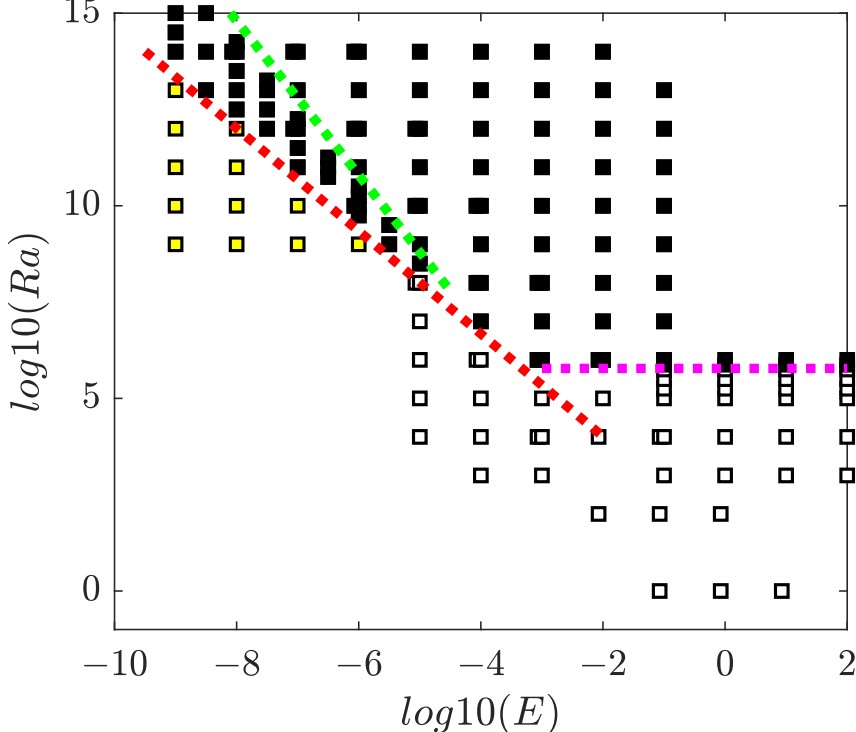

**Figure 4.** Parameter space covered by our log-lattice simulations at Pr = 0.7. The color of the symbols indicates the three possible regimes: conductive (white), transitional (yellow), and convective (black). The red line is the theoretical asymptotic prediction given by Equation (18). The magenta line is the theoretical convection threshold in the absence of rotation Ra = $6 \times 10^5$. The green line has equation Ra = $0.06E^{-2}$ and delineates regions of the parameter space where the turbulence is influenced by rotation (below the line) or not influenced by rotation (above the line), as diagnosed by the behavior of the kinetic energy dissipation, see Figure 5. The geostrophic turbulent regime is observed in between the red and the green line.

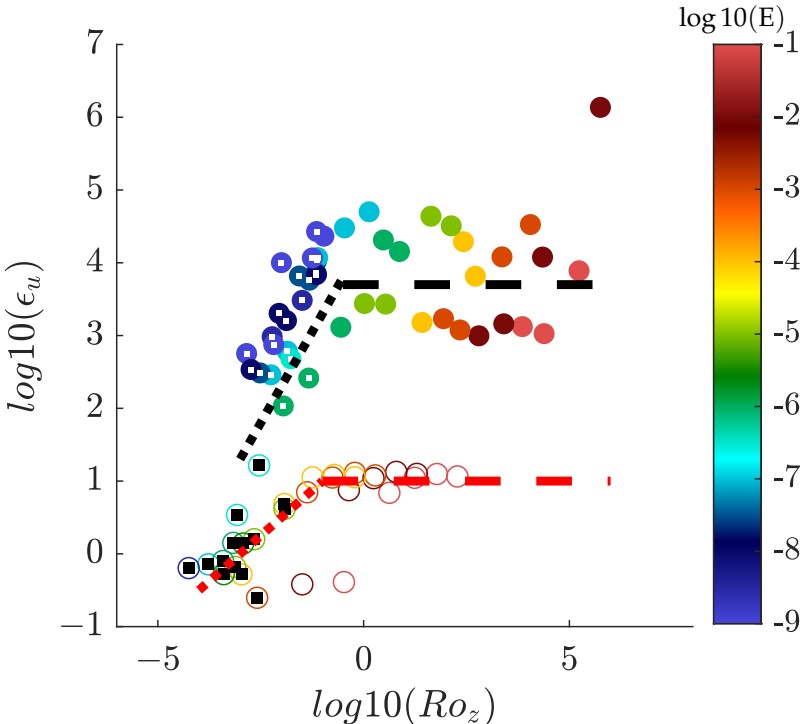

**Figure 5.** Non-dimensional energy dissipation $\epsilon_u = \nu < (\nabla u)^2 > H/U^3 = \mathrm{Ra}\,\mathrm{Nu}\,\mathrm{Pr}^{-2}/(\mathrm{Ra}\,\mathrm{Pr})^{3/2}$ vs. vertical Rossby number $\mathrm{Ro}_z$ in 3D for $\mathrm{Pr} = 0.7$ for rotating HRB simulations on log-lattices. The symbols are colored according to their Ekman number, E. The open symbols trace the laminar regime, while the filled symbols trace the turbulent regime. The rotation-dominated regimes are highlighted by a black (respectively, white) square for the laminar (respectively, turbulent) regime. The black (respectively, red) dotted line corresponds to $\epsilon_u \sim \mathrm{Ro}_z$ (respectively, $\epsilon_u \sim \mathrm{Ro}_z^{1/2}$), while the black (respectively, red) dashed line corresponds to $\epsilon_u = 3.7$ (respectively, $\epsilon_u = 1$).

### 5.3. Influence of Friction

To elucidate this observation, we computed the ratio of the kinetic energy dissipated by friction, $FU^2$, to the kinetic energy dissipated by in the flow by viscous processes, $\epsilon_u$. This is reported in Figure 6. We indeed observe two regimes: one at a low value of $\mathrm{Ra} < 10^{11}$, where the friction dominates the dissipation, and one at a larger $\mathrm{Ra} > 10^{11}$, where the friction is negligible. We will see below that this induces two different regimes termed "laminar" and "turbulent" regimes. In addition, we observe that as the rotation becomes larger, the influence of friction decreases.

### 5.3.1. Laminar vs. Turbulent Regime

Figure 7 reports the heat transport Nu as a function of Rayleigh number, Ra, and the Ekman number, E. One can see that larger Ekman numbers correspond to a smaller heat transport at a given Ra. However, rotation does not suppress the existence of a laminar to turbulent transition, which is already present in the non-rotating case. Like in the non-rotating case, it occurs when the vertical Reynolds number exceeds $u_z^{rms} L/\nu = \mathrm{Re}_c = 10^4$, corresponding to $\mathrm{Ra} \sim 10^{12}$, with a bi-stability occurring around $\mathrm{Ra} = 10^{11}$, see Figure 8. This critical value separates two different scaling regimes: one such that $\mathrm{Re} \sim \mathrm{Ra}^{1/2}$, and one such that $\mathrm{Re} \sim \mathrm{Ra}$.

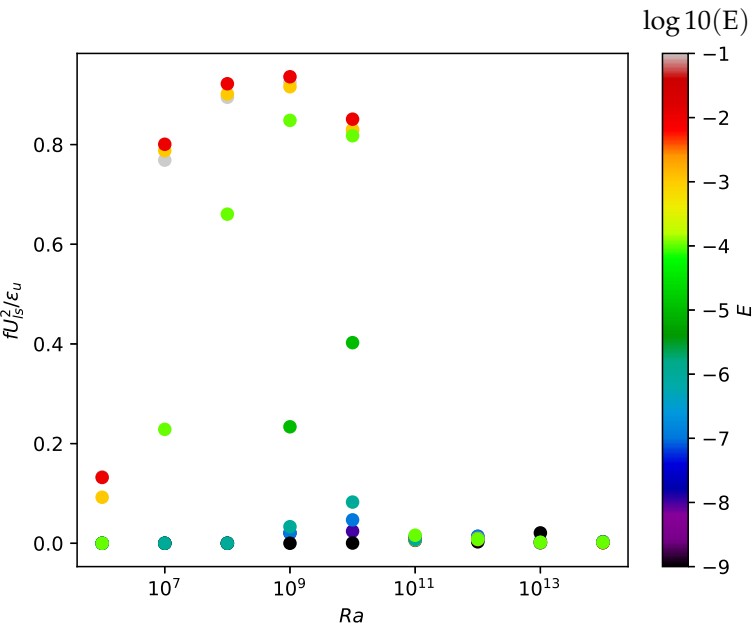

**Figure 6.** Ratio of the kinetic energy dissipated by friction, $FU^2$, to the kinetic energy dissipated by viscous processes, $\epsilon_u$, as a function of the Rayleigh number, Ra, and the Ekman number E. The points are color-coded by $\log 10(E)$. The friction dominates for low-rotation and for Ra $< 10^{11}$.

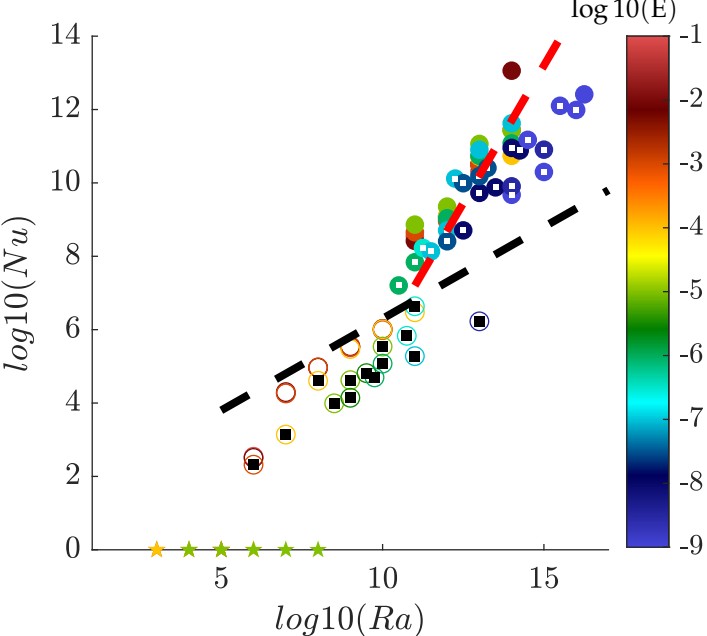

**Figure 7.** Non-dimensional heat transfer Nu vs. Rayleigh number Ra in 3D for Pr $= 0.7$ for rotating HRB simulations on log-lattices. The symbols are colored according to their Ekman number E. The stars traces the conductive regime. The open symbols trace the laminar regime, while the filled symbols trace the turbulent regime. The rotation-dominated regimes are highlighted by a black (respectively, white) square for the laminar (respectively, turbulent) regime. The black dashed line is Nu $= 20\sqrt{\text{Ra}}$, corresponding to asymptotic non-rotating ultimate regime scaling. The red dotted line is Nu $\sim$ Ra$^{3/2}$, corresponding to the geostrophic turbulent regime, see Figure 9 for an exact representation of the corresponding scaling law.

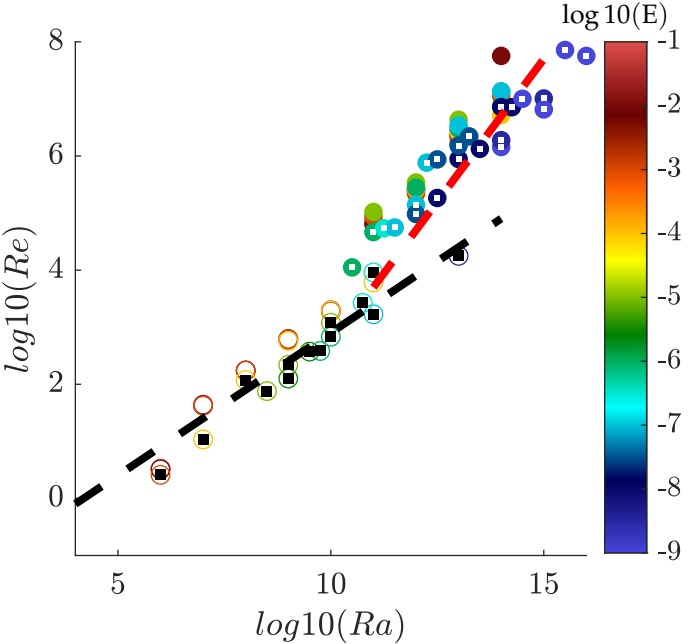

**Figure 8.** Vertical Reynolds number Re $= \sqrt{<u_z^2>}H/\nu$ as a function of Rayleigh number Ra in 3D for Pr $= 0.7$ for rotating HRB simulations on log-lattices. The symbols are colored according to their Ekman number, E. The open symbols trace the laminar regime, while the filled symbols trace the turbulent regime. The rotation-dominated regimes are highlighted by a black (respectively, white) square for the laminar (respectively, turbulent) regime. The black (respectively, red) dashed line follows the equation $y \sim x^{1/2}$ (respectively, $y \sim x^1$).

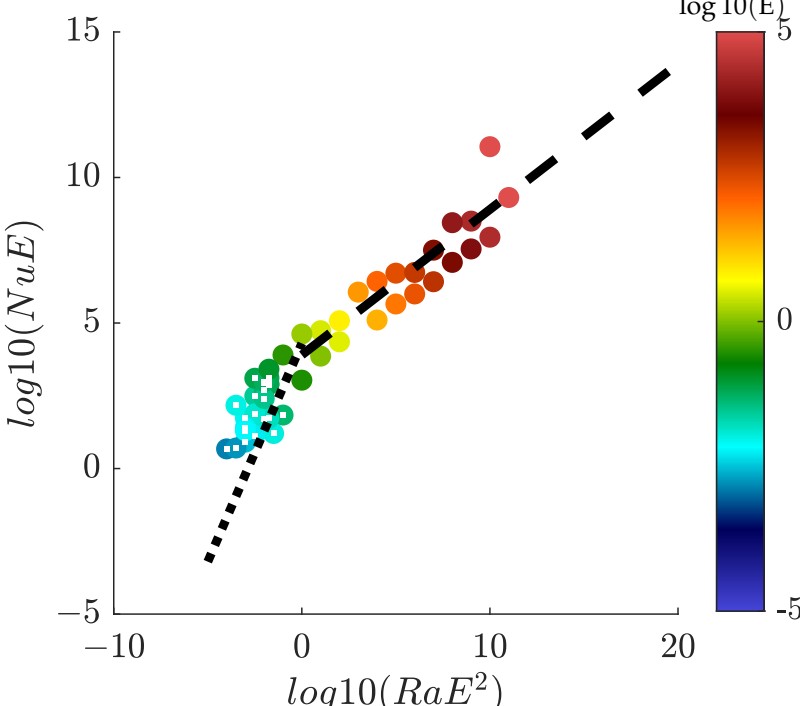

**Figure 9.** Universal law governing the heat transfer in the turbulent regime Nu E as a function of Ra E$^2$ in 3D for Pr $= 0.7$ for rotating HRB simulations on log-lattices. The symbols are colored according to their Ekman number E. The symbols tagged by a white square trace the rotation-dominated regime, while the filled symbols trace the rotation-independent regime. The black dotted (respectively, dashed) line follows the equation $y \sim x^{3/2}$ (respectively, $y \sim x^{1/2}$).

5.3.2. Influence of Rotation and Onset of Rotation-Dominated Regimes

The influence of rotation on turbulence is well documented (see, e.g., ref. [27,28]), and is characterized by a *decrease* in the efficiency of the transport properties with respect to the non-rotating case. For the case of heat transport, this is already clear from Figure 7, as already discussed. In the case of the turbulent kinetic energy dissipation $\nu < (\nabla u)^2 >$, it is known that the decrease is proportional to the vertical Rossby number, $\mathrm{Ro}_z = u_z/2H\Omega$, when $\mathrm{Ro}_z$ goes to zero [28]. We indeed observe this effect in our simulations, as illustrated in Figure 5, where the non-dimensional turbulent kinetic energy dissipation $\epsilon_u = \nu < (\nabla u)^2 > H/U^3$ is shown as a function of the vertical Rossby number. We see that in both the laminar and turbulent regimes, the energy dissipation indeed decreases for sufficiently low vertical Rossby numbers. In the turbulent regime, the decrease is indeed proportional to $\mathrm{Ro}_z$, indicated by the black dotted line. In the laminar regime, however, the decrease is milder, following $\mathrm{Ro}_z^{1/2}$. In the turbulent case, the rotation-dominated regime starts below $\mathrm{Ro}_z \sim 0.1$, while in the laminar case, it starts at $\mathrm{Ro}_z \sim 0.03$. We can use this change in regime to tag the simulations that are or are not influenced by the rotation. We denote them below by a white dot inside a filled symbol for the turbulent regime, and a black dot inside an open symbol for the laminar regime. Reporting this in Figures 7 and 8, we see that this regime corresponds to a lower heat transport and a smaller vertical Reynolds number, in agreement with the general findings that rotation impedes heat transport and vertical velocities. Note that the vertical Rossby number scales similar to the global Rossby number, following $\mathrm{Ro}_z \sim \mathrm{Ra}^{1/2}\, E$ in both the laminar and turbulent regimes, see Figure 10.

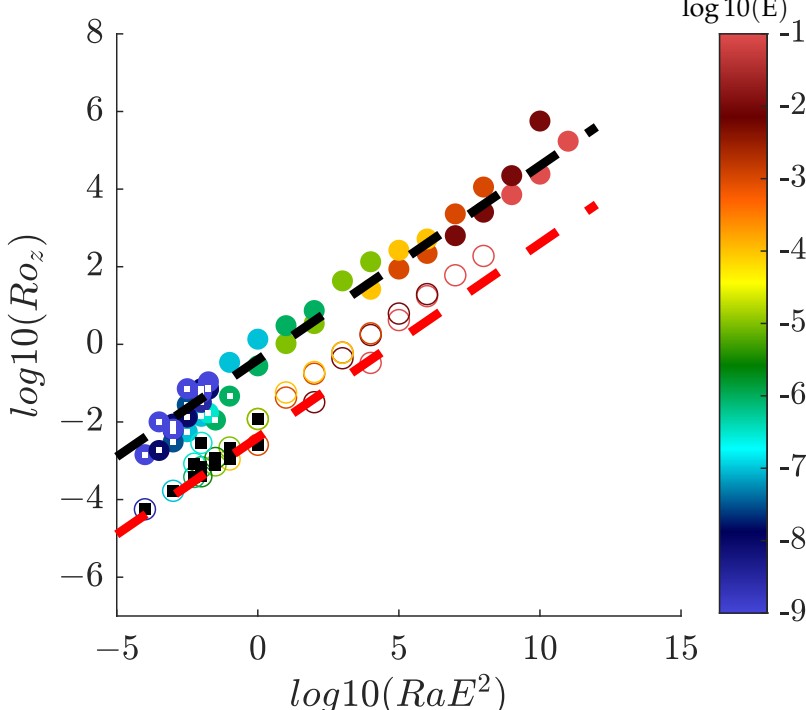

**Figure 10.** Vertical Rossby number $\mathrm{Ro}_z$ vs. "turbulent" Rayleigh number $\mathrm{Ra}\, E^2$ in 3D for $\mathrm{Pr} = 0.7$ for rotating HRB simulations on log-lattices. The symbols are colored according to their Ekman number $E$. The open symbols trace the laminar regime, while the filled symbols trace the turbulent regime. The rotation-dominated regimes are tagged by a black (respectively, white) square for the laminar (respectively, turbulent) regime. The black (respectively, red) dashed line follows $0.4\,\mathrm{Ra}^{1/2}\, E$ (respectively, $0.04\,\mathrm{Ra}^{1/2}\, E$).

### 5.3.3. Temperature Fluctuation and Anisotropy

Another interesting indicator of the influence of rotation on convection is given by the behavior of the temperature fluctuations, displayed in Figure 11. In the laminar case, they plateau at a low value (less than 0.01) as Ra increases, showing that the dynamics are indeed laminar. The rotation even tends to decrease the size of the fluctuations. In the laminar case, the temperature fluctuations increase with Ra, showing that convection is increasingly vigorous; even so, large rotations tend to somehow impede the development of excessively large fluctuations. The rotation also influences the anisotropy of the turbulence, as seen in Figure 12. Both in the laminar and turbulent cases, the anisotropy is well above 1/2, which would correspond to a situation where the kinetic energy is split evenly between motions along and perpendicular to the rotation axis. Due to the special nature of our projection, we cannot obtain a meaningful representation of what this result means in the physical space. It might, however, be interpreted as the influence of strong up (respectively, down)-drafts that are observed in direct numerical simulations to carry the heat from the bottom to the top (respectively, the cold fluid from the top to the bottom). The values we obtain here for the anisotropy are much larger to what are usually observed in simulations or experiments of RB convection with boundaries, but they are compatible with observations of radiative convection [29] or convection with no-slip boundary conditions [25], which have observed the formation of extremely extended plumes extending towards the whole bulk of the flow. The decrease in anisotropy observed at decreasing E values may be connected with the stabilizing influence of rotation, which impedes vertical fluctuations [27].

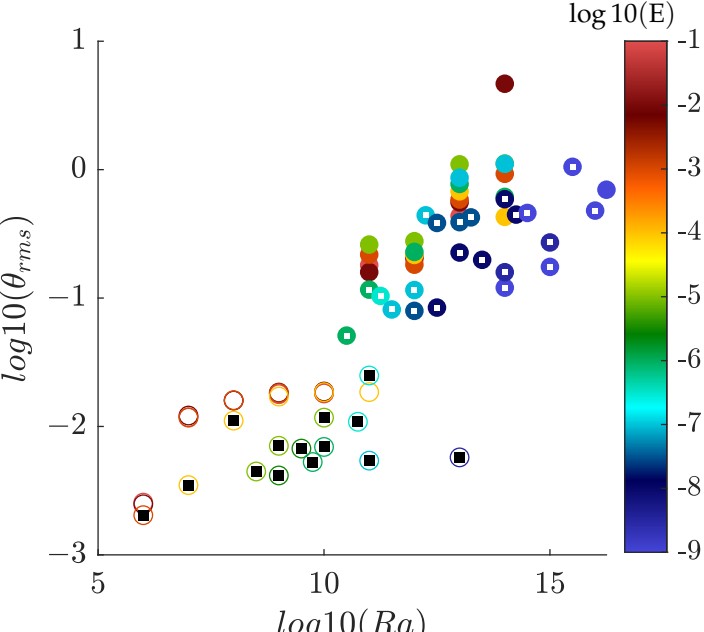

**Figure 11.** Temperature fluctuations $\theta_{rms} = \sqrt{<\theta^2>}$ vs. Rayleigh number Ra in 3D for Pr = 0.7 for rotating HRB simulations on log-lattices. The symbols are colored according to their Ekman number E. The open symbols trace the laminar regime, while the filled symbols trace the turbulent regime. The rotation-dominated regimes are highlighted by a black (respectively, white) square for the laminar (respectively, turbulent) regime.

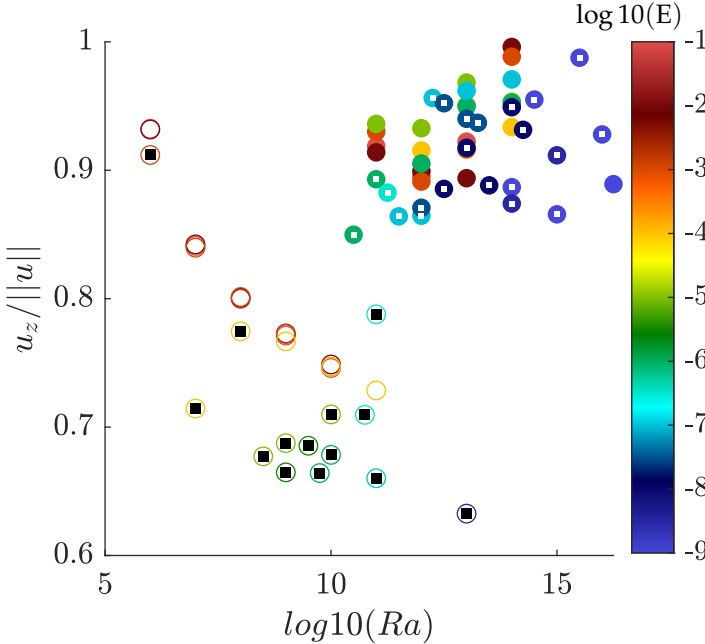

**Figure 12.** Velocity anisotropy $\sqrt{<u_z^2>}/\sqrt{<u^2>}$ vs. Rayleigh number Ra in 3D for Pr = 0.7 for rotating HRB simulations on log-lattices. The symbols are colored according to their Ekman number E. The open symbols trace the laminar regime, while the filled symbols trace the turbulent regime. The rotation-dominated regimes are highlighted by a black (respectively, white) square for the laminar (respectively, turbulent) regime.

5.3.4. Laminar and Turbulent Scaling Laws and GT Regimes

The laminar regime starts from the convection onset. It is therefore likely that it is influenced by near-onset dynamics. A natural idea is then to try to see whether it also follows the near-onset scaling law of the non-rotating case, Equation (22), albeit with $\mathrm{Ra}_c$ being replaced by its rotating value ($22\,\mathrm{E}^{-4/3}$) and $\mathrm{Ra}_t$ being replaced by $B\,\mathrm{E}^{-4/3}$, where the constant $B$ needs to be determined. With this hypothesis, we then find that in the laminar regime, Nu $\mathrm{E}^{2/3}$ should be a function of Ra $\mathrm{E}^{4/3}$, where the function satisfies Equation (22), with $A = 7$, $\mathrm{Ra}_c = 22$, and $\mathrm{Ra}_t = 5 \times 10^2$, see Figure 13. The turbulent regime corresponds to large Reynolds numbers, in which viscosity and diffusivity should not play a role any more. It is then natural to represent them in the turbulent variables Nu E and Ra $\mathrm{E}^2$ (where we have omitted the Pr dependence, since it a constant in all our data sets), which is shown in Figure 9. This representation indeed collapses the data with two different scaling laws: one with exponent 3/2 for regimes influenced by rotation—this is the GT regime—and one with exponent 1/2 corresponding to the turbulent regime not influenced by rotation. The precise location in the parameter space where the GT regime occurs can be computed using the fit of the vertical Rossby number as a function of Ra $\mathrm{E}^2$, see Figure 10. The condition $\mathrm{Ro}_z < 0.1$ then translates into the condition Ra $\leq 0.06E^{-2}$, which is the green line in Figure 4. The GT regime is then found to be in-between the blue and the green lines, which is the region where we concentrate additional numerical simulations.

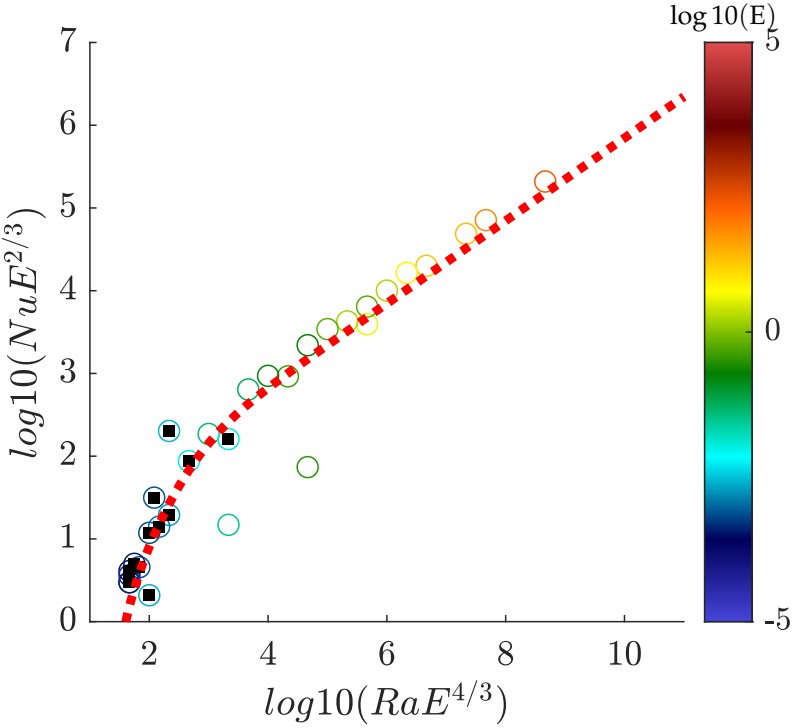

**Figure 13.** Universal law governing the heat transfer in the laminar regime $Nu\,E^{2/3}$ as a function of $Ra\,E^{4/3}$ in 3D for $Pr = 0.7$ for rotating HRB simulations on log-lattices. The symbols are colored according to their Ekman number E. The open symbols containing by a black square trace the rotation-dominated regime, while the open symbols trace the rotation-independent regime. The red dotted line follows Equation (22), with $A = 7$, $Ra_c = 22$ and $Ra_t = 5 \times 10^2$.

## 6. Discussion

We have shown that the projection of geophysical equations of motion allow for the determination of realistic values of parameters at a moderate cost. This allows us to perform many simulations over a wide range of parameters, thereby leading to general scaling laws of transport efficiency that can then be used to parametrize the turbulent transport in general climate models for Earth or other planets. We have illustrated this process using the equation describing the heat transport in a dry atmosphere to obtain the scaling laws for the onset of convection as a function of rotation, and confirmed the theoretical result of $Ra_c \sim E^{-4/3}$ over a wide range of parameters. We have also demonstrated the existence of two regimes of convection: one laminar regime extending near the convection onset and one turbulent regime occurring as soon as the vertical Reynolds number reaches a value of $10^4$. We have derived general scaling laws for these two regimes, both for the transport of heat and the dissipation of kinetic energy, and values of the anisotropy and temperature fluctuations. The set-up we have used here is far from being able to reproduce the full complexity of the atmosphere, as it models the friction at the bottom with a simple law, and ignores moisture dynamics. We plan to include these features in future work. Finally, it is not clear how the projection of dynamics onto log-lattices influences the results we are deriving. It is quite remarkable that the procedure is able to capture scaling laws, as we recover here some results already obtained in experiments [29] and numerical simulations [25] of convection without boundary layers, albeit with different prefactors. This of course has some important implications when translating these scaling laws as parametrizations in models. However, if we believe that the scaling laws themselves are robust, it only takes comparisons with a few direct numerical simulations to recalibrate the constants and turn our laws into useful parametrizations.

**Author Contributions:** Conceptualization, all authors; methodology, all authors; software, A.B., G.C. and Q.P.; validation, all authors; formal analysis, B.D.; investigation, Q.P.; resources, A.B.; data curation, all authors; writing—original draft preparation, B.D.; writing—review and editing, all authors; visualization, B.D.; supervision, B.D.; project administration, B.D.; funding acquisition, B.D. All authors have read and agreed to the published version of the manuscript.

**Funding:** This work received funding through the PhD fellowship programs of the Ecole Polytechnique and Ecole Normale Superieure Paris-Saclay, and through the Agence Nationale pour la Recherche, via the grants ANR TILT grant agreement no. ANR-20-CE30-0035 and ANR BANG grant agreement no. ANR-22-CE30-0025.

**Institutional Review Board Statement:** Not applicable.

**Informed Consent Statement:** Not applicable.

**Data Availability Statement:** The data presented in this study are available on request from the corresponding author. The data are not publicly available due to restrictions.

**Conflicts of Interest:** The authors declare no conflict of interest.

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
