# Peer review of "Log-Lattices for Atmospheric Flows"

_atmosphere, doi:10.3390/atmos14111690_

Round 1
Reviewer 1 Report
Comments and Suggestions for Authors
This manuscript presents numerical simulations of rotating convective turbulence using the log-lattice approach to reach very high Reynolds numbers. In this approach, which is an alternative to large eddy simulation, mode reduction is attained by sampling/averaging wavenumbers logarithmically across all scales. The topic is very appropriate for this special volume, since the method and application are both topics that were of great interest to Jack Herring. The method is promising and well presented, and the results are very interesting. Overall I think this is a really nice and well written manuscript. I just have a few comments and suggestions:
Lines 86 and 91, what is the "}"?
What is the purpose of averaging the KE spectrum on the logarithmically spaced wavenumbers? (Equation 4). Does this model include averaging over these wavenumber intervals (and therefore incorporate extra term, analogous to sub-filter-scale fluxes in LES), or are intermediate wavenumbers between the lambda^m simply truncated/neglected?
What is M in (4), (5), and (6)?
Line 131, I don't think you need the "+\theta" since this is the equilibrium profile.
Section 4.4 does not seem specific to the log-lattice approach and should probably include more reference to rotating convective instabilities, unless these are original results.
How are the simulations done? Is the convolution sum computed directly or are FFTs used?
Lines 227-230, I have lost track of what N is.
How sensitive are these results to lambda?
Comments on the Quality of English Language
There are just some minor typos and unusual word choices, e.g.:
Line 76, should prize be price?
Line 133, does "sequel" mean below or in a follow-up paper?
Author Response
Referee 1
This manuscript presents numerical simulations of rotating convective turbulence using the log-lattice approach to reach very high Reynolds numbers. In this approach, which is an alternative to large eddy simulation, mode reduction is attained by sampling/averaging wavenumbers logarithmically across all scales. The topic is very appropriate for this special volume, since the method and application are both topics that were of great interest to Jack Herring. The method is promising and well presented, and the results are very interesting. Overall I think this is a really nice and well written manuscript. I just have a few comments and suggestions:
Lines 86 and 91, what is the "}"?
It is a typo. Corrected.
What is the purpose of averaging the KE spectrum on the logarithmically spaced wavenumbers? (Equation 4). Does this model include averaging over these wavenumber intervals (and therefore incorporate extra term, analogous to sub-filter-scale fluxes in LES), or are intermediate wavenumbers between the lambda^m simply truncated/neglected?
Because the wavenumbers follow a geometric progression (equation 2), the number of wavenumbers in a thin spherical layer does not evolve like 4 pi k^2 dk, but is much lower. If we computed the KE spectrum following an arithmetic progression of the layers, there would be layers at high k where no points are present at all. The density of points evolves like (log k)^2 / k^3, and is not at all smooth between lambda^m and lambda^(m+1). Having a KE spectrum defined in coherence with the geometric progression of modes is natural.
This model does not include averaging over wavenumbers. As a REWA model, many wavenumbers are not computed at all.
What is M in (4), (5), and (6)?
It is a typo. We fixed it.
Line 131, I don't think you need the "+\theta" since this is the equilibrium profile.
T is the temperature, and the equilibrium profile is T = Teq = -Delta T z / H, corresponding to theta = 0. What is put line 131 is the definition of theta.
Section 4.4 does not seem specific to the log-lattice approach and should probably include more reference to rotating convective instabilities, unless these are original results.
These are traditional results adapted to the log-lattice case. The eq. 184 P.106 chap. III of the book “Hydrodynamic and Hydromagnetic Stability” by Chandrasekhar is similar to our equation 18. This reference has been added.
How are the simulations done? Is the convolution sum computed directly or are FFTs used?
The convolution sum is computed directly, but it is less costly than a traditional FFT because only some local interactions are kept.
Lines 227-230, I have lost track of what N is.
N is defined such that the maximum wavenumber in the x, y or z direction is prop. to $ \lambda^N$. We have recalled this at the beginning of the paragraph.
How sensitive are these results to lambda?
Lambda = 2 is ill suited in the incompressible case because a lot of term of the convolution cancel each other. Lambda = 1.32 was not much tested in the present configuration, because it has more computational cost. However, in simpler configurations with no stratification and no rotations, we checked that scaling laws are insensitive to $\lambda$. We mentioned this in a new paragraph.
Comments on the Quality of English Language
There are just some minor typos and unusual word choices, e.g.:
Line 76, should prize be price?
Corrected.
Line 133, does "sequel" mean below or in a follow-up paper?
It means below. We corrected it.

Reviewer 2 Report
Comments and Suggestions for Authors
This is a welcomed investigation of rotating convective turbulence using the log-lattice model. I willing to recommend publication of the article provided the authors address the following issues:
1) There is a lot of discussion of what is captured correctly by the model but no discussion about what is not. I would like thus that the authors discuss the limitations of the model and what it does not capture. Among others, the effect of periodic boundary conditions used, the absence of boundary layers, absence of intermittency, the effect of the large scale drag etc.
2) Some of the plots are not clear, convey no message or can be improved.
a) Fig 4. can be improved.
(i) Why use so thick symbols? there is overlap in the QG region.
(ii) Why aren't there points of log(E)>0 to demonstrate the validity of the pink line?
(iii) I did not see the point in distinguishing yellow from white points both decay.
How do the authors distinguish QG points from turbulent (green line)?
b) Fig. 5 is there a bistablity at log(Ra)=6 or is this an effect of different E ploted?
c) Fig 6 what is the power law shown in this figure? There is no prediction in this case to draw a line?
d) Fig 9 & 10 is there something to be concluded from these figures? Can they be omitted? I see no trend.
Reviewer 3 Report
Comments and Suggestions for Authors
Please see attached PDF file.

Author Response
I understand that this paper is being considered for publication as part of a collection
of papers in honor of the late Dr. Jack Herring. Like the senior author of paper,
I have held Dr. Herring in high regard and have benefitted much from his kindness
in my earlier career development.
The focus on the paper is on the premise of a novel spectral numerical scheme
that can allow ultra-high Reynolds numbers to be achieved, at a cost is that many,
many orders of magnitude lower than the largest simulations possible on the most
power supercomputers available (or even decades later). The trick is apparently to
simulate the turbulence only at a very modest number of Fourier modes which are
logarithmically rather than arithmetically spaced on the wavenumber axis. If I understand
correctly this is essentially a fully-spectral technique where Fourier coefficients
of nonlinear terms are obtained as convolution sums — but only using a very small
number of wavenumber modes. The authors present results on scaling laws such as
k−5/3 energy spectrum in Figure 1 which seem realistic despite the unconventional
nature of the proposed technique.
My thinking is that, while this technique is very interesting, it is also not perfect or
not complete. Most scaling laws in this paper are based on second-moment quantities.
They just do not give a complete picture of the turbulence — for instance what
about intermittency, higher-order moments, etc. The representation of the spectrum
E(k) by just 20 discrete wavenumbers presumably makes quantities that depend on
derivatives such as dE(k)/dk very uncertain. I think the authors should acknowledge
that the new technique does not provide a complete description of the turbulence,
and does not render the pursuit of high-performance computing in turbulence research
any less important.
The referee is of course right. There are various limitations to this technique, the most important being impossibility to describe non-local interactions, lack of intermittency, impossibility to capture resonance for dispersive waves, ..
We have added several paragraphs:
Computations on log-lattices involve a number of limitations. First, due to the local nature of the convolution in Fourier space, log-lattice are unable to describe the so-called "non-local" interactions that are involved for example in the shearing of small eddies by large eddies \cite{LDN01}. This may explain why lo-lattices simulation do not display intermittency for the structure functions \cite{CBD23}. Due to their sparsity, log-lattices cannot describe dispersive wave resonances, like inertial or gravity waves. This mans that they cannot capture dissipative phenomena induces by those waves, and that it can only capture the dissipation due to small scale turbulent eddies.
Finally, due to the spectral nature of the construction, it may appear that the log-lattice framework is only appropriate for homogeneous flows, i.e.\ far from boundaries.
In a recent work,~\cite{campolina22} have however shown that the extension to flow with boundaries is possible, via lattice symmetrization around the boundary and careful treatment of the resulting discontinuity.
Despite the high relevance of this situation for geophysical flow, we here concentrate on the simpler case of homogenous flow, and show how log-lattice simulation enable to recover some well known features of homogeneous rotating convection.
Another limitation concerns the lack of k_i=0 mode. This means that “vertical” structures observed in rotating turbulence can’t be described in purely vertical terms [k_z=0] .
While the issue above is my main point, I also think that some of the graphics
are not easy to interpret. Take for instance Fi 4. The x-axis has 12 decades while
the y-axis has 15. A 100% i.e. a factor of 2 discrepancy is hardly noticeable. But
the symbols are supersized: not much smaller in one decade on the scales being used.
We have redone some figures with smaller data points.
Some of the figure captions are show substantial scatter, as well. The meanings of
phrases such as “black (resp. white)” and something (resp. something else) are also
unclear to me. (I do now know what “resp.” stands for in this context.)
It means respectively. We have put the whole word
Overall, I have no problem seeing this paper published but I also think the points
above need attention. (In particular, this approach does not provide the full story,
and its limitations need to be acknowledged, too.)
